# Detection of Debonding Defects in Carbon Fiber-Reinforced Polymer (CFRP)–Rubber Bonded Structures Based on Active Lamb Wave Energy Analysis

**DOI:** 10.3390/s24175567

**Published:** 2024-08-28

**Authors:** Zhenze Yang, Yongfeng Ren, Qiang Shi, Dapeng Cui, Jieqing Liu

**Affiliations:** State Key Laboratory of Electronic Test Technology, North University of China, Taiyuan 030051, China; 15735116460@163.com (Z.Y.); shiqiang0221@163.com (Q.S.); b20230601@st.nuc.edu.cn (D.C.); sz202206130@st.nuc.edu.cn (J.L.)

**Keywords:** solid rocket motor, nondestructive testing, structural health monitoring, interface debonding detection, Lamb waves, CFRP–rubber composite, PZT

## Abstract

Carbon fiber-reinforced polymers (CFRPs) are widely used in the fabrication of solid rocket motor casings due to their exceptional performance. However, the bonding interface between CFRP and viscoelastic materials (rubber) is prone to debonding damage during service and storage under complex environmental conditions, which poses a significant threat to the structural integrity and reliability of the engine. Existing nondestructive testing (NDT) methods, such as X-ray imaging, infrared thermography, and ultrasonic testing, although somewhat effective, exhibit significant limitations in detecting interfacial defects in deep or multilayered composite materials, particularly under the challenging conditions of service and storage. This study proposes an innovative method based on active Lamb wave energy analysis and introduces the Damage Evolution Factor (DEF), specifically designed to detect and evaluate interfacial debonding defects in CFRP–rubber bonded structures within solid rocket motors during service and storage. Through numerical simulations and experimental validation, we selected the A0 mode Lamb wave, which is more sensitive to interfacial damage, as the incident wave and excited it on the surface of the structure. Displacement time-history response signals at observation points under different damage models were extracted and analyzed, and DEF values were calculated. The results show that DEF values increase with the size of the interfacial debonding damage. Similar trends were observed in experimental studies, further validating the effectiveness of this method and demonstrating that DEF can be used for the quantitative evaluation of interfacial debonding defects in CFRP–rubber bilayer bonded structures.

## 1. Introduction

Solid rocket motors (SRMs) are extensively utilized in aerospace and missile systems because of their simple structure, high thrust, mobility, reliability, and ease of maintenance [1]. To reduce the weight and increase the operational pressure of SRMs, traditional metallic materials in motor casings have been increasingly replaced with carbon fiber-reinforced polymer (CFRP) composites owing to their superior mechanical properties [2,3,4]. The structural configuration of a typical SRM, as illustrated in Figure 1, includes a CFRP shell, an insulation layer, a liner, and a solid propellant from the outside to the inside [5].

However, during their service and storage, SRMs are exposed to various loads, such as different temperatures and vibrations, which can damage the structural integrity of these motors. Interfacial debonding damage, which tends to occur at the interface of the propellant, liner, insulation, and shell, can lead to severe consequences such as burning instability, thrust loss, and in extreme cases, catastrophic structural failure. These damages not only compromise the operational performance but also pose significant risks to mission success and safety. In particular, approximately one-third of solid motor launch failures are caused by interfacial debonding [6,7,8]. Therefore, structural health assessment of SRMs during their service and storage periods is essential [5,8,9].

In recent years, numerous nondestructive testing (NDT) methods, including X-ray imaging, infrared thermography, and ultrasonic testing, have been developed to detect structural damages in SRMs.

Wei et al. [10] utilized real-time X-ray imaging to measure the burning rate of solid propellant within the combustion chamber of an SRM, providing valuable insights into the operational pressures experienced during combustion. Hoffmann et al. [11] developed an improved algorithm based on X-ray techniques to detect surface defects on SRM liners, effectively enhancing the detection process. Similarly, Mini et al. [12] proposed an X-ray-based method for detecting debonding damage by analyzing the structural shape of SRMs, identifying potential debonding locations with high accuracy. Thus, this proposed X-ray-based method represented an optimal approach for detecting interfacial debonding. Li et al. [13] proposed an algorithm for detecting defects in films using a high-resolution X-ray-based technique, HDCAM, and a cross-layer feature fusion attention mechanism. This proposed algorithm outperformed the existing state-of-the-art algorithmic models and yielded the best detection results.

Moreover, several researchers have employed infrared thermography for debonding detection in SRMs. Liu et al. [14] investigated SMR debonding defects using infrared thermography. Wang et al. [15] used multimodal optical excitation pulse thermography to detect debonding defects at solid rocket motor propellant/cladding interfaces. This multifeature extraction algorithm improved the detection efficiency and dynamic range of depth resolution of solid rocket motor interface debonding defects. Liu et al. [16] proposed a depth autoencoder thermography method for detecting subsurface defects in composites. In addition, a new quantitative metric was proposed to compare the defect detection performances of different methods. Cholevas et al. [17] significantly improved the detection performance of a structural health monitoring system using the Neyman–Pearson detection theory to differentiate the damage information of SRM structures collected by embedded strain sensors.

Guo et al. [18] classified stripping defects, detected in SRMs using ultrasonic nondestructive testing techniques, with the aid of a model combining wavelet packet transform and machine learning; this new detection technique was suitable for structural health characterization of SRMs.

Despite the progress made in nondestructive testing techniques such as X-ray imaging, infrared thermography, and ultrasonic testing, these methods face significant challenges in field applications. For instance, X-ray imaging, while capable of producing high-resolution internal images, requires complex equipment and stringent safety precautions. Moreover, the inspection process typically involves disassembling components of solid rocket motors (SRMs), leading to increased time and labor costs, making this approach impractical in operational environments. Similarly, infrared thermography can detect surface and near-surface defects without contact, but its effectiveness is limited for detecting deep-seated flaws, especially in multilayer composite structures. Embedded sensor techniques, though offering real-time monitoring, are also costly and complex, limiting their widespread application in engineering practice.

In contrast, ultrasonic guided wave methods, especially Lamb wave NDT methods based on lead zirconate titanate (PZT) transducers, offer unique advantages in overcoming these limitations. Lamb waves can travel long distances within complex structures and are highly sensitive to changes in material properties, making them well-suited for addressing the limitations of X-ray and infrared thermography, especially in detecting deep and complex structural defects [19,20]. Unlike traditional methods, Lamb wave-based techniques do not require complex equipment or disassembly, significantly reducing the time and safety requirements for field inspections. Furthermore, the ability of Lamb waves to provide extensive coverage and access to hidden structures in harsh environments positions them as an ideal choice for structural health assessment of SRMs. Thus, Lamb wave methods not only offer significant advantages in addressing the shortcomings of existing technologies but also present a highly promising solution for detecting interfacial debonding defects in SRMs.

In recent years, significant progress has been made in the research on detecting interfacial debonding damage using Lamb waves. For instance, Li et al. [21] studied interfacial debonding by attaching PZT patches to the surface of concrete-filled steel tubular (CFST) members. Their results showed that the output voltage amplitude increased significantly with the size of the debonding area. Wang et al. [22] employed a wave-based method using piezoelectric ceramics to successfully evaluate the peeling damage at the CFRP–concrete interface by analyzing the amplitude of sinusoidal signals and the wavelet packet energy of frequency sweep signals. Nitesh et al. [23] applied a nonlinear Lamb wave method to detect delamination damage in composite laminates, demonstrating the high sensitivity of nonlinear Lamb waves in identifying minor damage. Li et al. [24] utilized anti-symmetric Lamb wave modes to detect debonding in CFRP–reinforced steel structures. The results indicated that as the debonding size increased, the time of arrival of the wave at the sensor significantly decreased, confirming the potential of this method in multilayer composite applications. Liu et al. [25] through numerical simulations and experiments, studied debonding in adhesive-bonded metal waveguides, finding that with an increase in the debonded area, the time of arrival of the guided wave modes significantly decreased, further validating the effectiveness of this method in detecting interfacial debonding in bonded structures.

Despite the important progress achieved in these studies, research on the theoretical aspects of interfacial debonding detection remains limited and requires further exploration. This study extends the work of Chen et al. [26] by not only considering Lamb wave energy leakage but also introducing the impact of energy loss at the site of interfacial damage. A Debonding Evaluation Factor (DEF) is defined, specifically for detecting and assessing interfacial bonding defects in CFRP–rubber (viscoelastic) bonded structures within solid rocket motors (SRMs) during operation and storage.

The paper is organized as follows: Section 2 outlines the detection principle, focusing on Lamb wave energy loss due to debonding, and introduces a novel interface debonding Damage Evaluation Factor (DEF). Section 3 describes the simulation and experimental methodologies. Section 4 presents the results, along with detailed analysis and discussion. Finally, Section 5 concludes with the key findings of the study.

## 2. Detection of Debonding Defects Based on the Energy Loss of Lamb Waves

### 2.1. Lamb Wave Energy Variations in Plate Bonded Structures

In a well-bonded layered medium, the Lamb wave energy decreases as the propagation distance increases, which can be attributed to two main factors [26]. First, energy is lost due to the inherent damping characteristics of structural materials, a phenomenon commonly referred to as energy dissipation. Second, the Lamb waves propagating in the plate medium propagate to the neighboring medium in contact with it and transfer the energy to it, resulting in the leakage of the Lamb wave energy. Thus, the Lamb wave energy dissipation in a plate media is assumed to be exponential, as follows [27]:(1)$$Q_{l}=Q_{0}e^{-2\alpha l}$$
where $Q_{0}$ is the acoustic intensity at the point of excitation, $Q_{l}$ is the acoustic intensity after the propagation distance l, and α is the energy dissipation factor of the Lamb wave.

The energy leakage from the Lamb waves is related to the contact area between the flat plate and neighboring media. Assuming that the area of the face in contact with the neighboring media is *d* × *l*, where *d* is the length and *l* is the contact width taken as 1 for convenience, energy leakage can be represented as [28]:(2)Qd=Q0ⅇ−2βd
where Qd is the acoustic intensity after energy leakage, and β is the energy leakage factor of the Lamb wave.

When debonding damage occurs in a structure, the Lamb wave energy is reflected/refracted at the damage boundary, whereas it undergoes scattering in the damaged area. The energy scattering in the damaged region is correlated to the debonding area in the structure. In this study, assuming that the area of debonding is x × 1 (where x is the length of the debonding surface, and 1 is the width of the debonding (unit width)), the loss of Lamb wave energy at the debonding damage can be expressed as:(3)Qx=Q0ⅇ−2γx
where Qx is the sound intensity after energy leakage in the debonding region, and γ is the energy loss factor in the debonding damage region.

The Lamb wave energy transmission is shown in Figure 2. Suppose that material A and B form a plate-like structure with debonding damage, the length of the bonding zone is l−x with point B as the left boundary of the debonding damage, whereby the distance to point A is $\left(l-x\right)/2$. As the Lamb wave spreads from the actuator (point A) to the sensor (point D), energy changes as follows: (1) energy dissipation in material A (black arrow); (2) energy leakage in the bonding region between material A and B (purple dotted arrows); (3) reflection/refraction of Lamb wave energy at the damage boundary (blue arrow); and (4) Lamb wave energy scattering in the debonding region of material A and material B (yellow arrow). In the debonding region, the Lamb wave energy no longer leaks energy into the B layer of the material.

The relationship in the sound intensity between actuators A (QA) and sensor D (QD) can be expressed as follows:(4)QD=QAⅇ−2α+βl+2β−γx

### 2.2. Principle of Interfacial Defect Detection Based on Lamb Wave Debonding Energy Loss

It is assumed that two plates consisting of different materials bond to each other at different interfacial debonding lengths, as shown in Figure 3. The separation between the actuator and sensor 2 is denoted as l, and the extent of interfacial debonding damage along the actuator–sensor trajectory is represented by x. As the Lamb wave transmits from the actuator to sensor 2, the energy dissipation remains the same. When interfacial debonding damage occurs between the actuator and sensor 2, leakage of Lamb wave energy from material A to material B occurs in the bonding region. When passing through the damaged region, the Lamb wave energy is reflected/refracted at the damaged boundary as well as scattered in the damaged region, and the energy in the damaged region does not leak to the B layer of the medium as shown in Figure 3c. With the enlargement of interfacial debonding defects, the Lamb wave energy transmitted to material B declines. The sequence of the signal energies received by sensor 1 is as follows: Q_1–3_ > Q_1–2_ > Q_1–1_. Meanwhile, the sequence of the signal energies received by sensor 2 is: Q_2–3_ > Q_2–2_ > Q_2–1_. Therefore, the interfacial defect detection based on the Lamb wave debonding energy loss can be used to detect the interfacial debonding damage in bonded structures.

### 2.3. Interface Debonding DEF

A wave-energy-based damage evaluation index was proposed to evaluate the extent of the interface debonding defects and quantitatively study its damage. According to the acoustic intensity equation, as shown in Equation (5), the correlation between the sound intensity and peeling length x can be expressed as follows:(5)QDQ0=ⅇ2β−γx
where x is the length of the debonding damage, and QD is the sound intensity when a length x in the structure undergoes interfacial debonding damage, and Q0 is the sound intensity without debonding.

The DEF for debonding at the interface of a CFRP–viscoelastic bonding material is defined as the logarithm of the ratio of the acoustic field strength to the detection path length, as shown in Equation (6):(6)$$DEF=\frac{1}{l}\ln\left(\frac{Q_{D}}{Q_{0}}\right)=2\left(\beta -\gamma \right)\frac{x}{l}$$

Given that the acoustic intensity is proportional to the acoustic energy, we can replace the acoustic intensity with the magnitude of the amplitude energy corresponding to the frequency of the excitation signal after Fourier transform. DEF can then be represented as:(7)$$DEF=\frac{1}{l}\ln\left(\frac{E_{D}}{E_{0}}\right)=2\left(\beta -\gamma \right)\frac{x}{l}$$
where l is the detection path length, $E_{D}$ is the amplitude energy corresponding to the frequency of the excitation signal after the Fourier transform of the signal when debonding defects occur at the interface of length x in the structure, and $E_{0}$ is the amplitude energy corresponding to the frequency of the excitation signal after the Fourier transform of the signal obtained in the absence of debonding defects in the structure.

Here, DEF is proportional to the debonding length. Once β and γ are determined, the degree of damage (x/l) can be evaluated quantitatively.

## 3. Methods and Results

### 3.1. Excitation Signal Selection

Owing to the effects of dielectric boundaries [29], Lamb waves have the inherent properties of multimodality, frequency dispersion, and attenuation, which significantly increase the difficulty of signal processing. Therefore, a suitable incident signal for a Lamb wave-based nondestructive testing method should be selected. Depending on the motion direction of particles, Lamb waves can be categorized into two modes: symmetric (S; e.g., S0, S1, and S2) and antisymmetric (A; e.g., A0, A1, and A2). In A waves, particles exhibit predominantly out-of-plane motion, suggesting that the asymmetric mode is more sensitive to interfacial debonding damage. Hence, the A mode was selected as the incident wave for detecting the interfacial debonding damage in the bonded structures.

Owing to the inherent frequency dispersion of Lamb waves, both their phase and group velocities are determined by the wave frequency (f), thickness of the material (h), and other factors. Typically, a signal of limited bandwidth containing a specific number of cycles serves to minimize dispersion. Therefore, a Hanning window covering five cycles was utilized to restrict the bandwidth of the chosen Lamb wave mode and thus enhance the signal-to-noise ratio, as shown below:(8)$$P\left(t\right)=\left\{\begin{array}{c} \frac{1}{2}*\left[1-cos(\frac{2\pi ft}{N})\right]\sin\left(2\pi ft\right),\;\;\;for\;0\leq t\leq \frac{N}{f}\\ 0,\;\;\;for\;\;\;t\geq \frac{N}{f} \end{array}\right.$$
where f is the center frequency of the excitation signal, N is the number of repetition cycles of the excitation signal, and t is the signal duration.

The velocities of the Lamb wave modes at various excitation frequencies and within diverse materials are typically extracted from dispersion curves, which are meticulously computed and graphed using the commercial software tool, “GUIGUW 2.2” [30]. The parameters of the selected materials are listed in Table 1, and the group velocity dispersion curves of the CFRP veneer, rubber veneer, and CFRP–rubber bonded sheets are shown in Figure 4. Note that the ordinate (group velocity) is usually expressed in C_g_.

In this study, the thickness and fiber orientation of each layer of CFRP in Table 1 are as follows: Each CFRP layer has a thickness of 0.5 mm, and the fiber orientation is [0°/90°/±45°] staggered, with a total of 4 layers. The average values shown in Table 1 are based on the overall average of the above layer structures and reflect the mechanical properties of the entire laminate structure. These values are not the properties of a single layer but take into account the superposition effect of a multilayer composite.

As shown in Figure 4, it is evident that only the fundamental modes, i.e., A0 and S0, are induced at an approximate frequency of 100 kHz. The group velocity changes in the A0 and S0 modes are relatively flat and different. Therefore, 100 kHz was considered as the appropriate frequency to excite the Lamb wave. In this investigation, stress waves were excited and detected along the circumference of the engine structure, which can be considered as circumferential guided waves [27]. The dispersion properties of circumferential guided waves are predominantly affected by the curvature and thickness of the structures [31]. When the curvature or thickness of the loop structure is sufficiently small, its dispersion curve can be approximated as a flat plate [32,33]. The dispersion curves of single-layer annular voids and flat plates of the same material and thickness were calculated and compared using the open-source software GUIGUW [30]. The dispersion curves of the ring–space and flat-plate structures are shown in Figure 5.

As shown in Figure 5, the dispersion characteristics of the guided waves in the ring–space plate correspond with those in the flat plate (whether in the S0 mode or in the A0 mode). Therefore, subsequent simulation modeling was performed using a flat-plate structure instead of an annular structure.

### 3.2. Finite Element Development

The propagation of Lamb waves in the CFRP–viscoelastic material bonded plates was simulated using the commercial finite element package ABAQUS 2022/Explicit. A CFRP sheet with an area of 450 mm × 100 mm and a thickness of 2 mm was considered. The viscoelastic material was replaced with rubber of the same dimensions as the CFRP sheet. The actuator was placed at the far-left end of the CFRP plate, and the sensor (observation point) at 325 mm, as shown in Figure 6. A five-cycle sinusoidal Hanning window modulated wave with an excitation frequency of 100 kHz was applied to the flat plate. The direction of the force was 90° to the CFRP plate, and the time-range response of the displacement at the sensor position was obtained as the output signal [26]. For simplicity, CFRP laminates were set as isotropic materials using the material parameters in Table 1 because the current study only focuses on wave propagation along the longitudinal direction of the CFRP laminate without considering the directional effect of the composite.

In the simulation, a “Tie” constraint was implemented to replicate the interplay between the CFRP sheet and rubber within the bonding region. The interaction in the debonding region was set to unravel. The model was meticulously meshed using a CPE4R cell, employing a maximum mesh size of 1 mm and a time step of 1 × 10^−7^ s.

#### 3.2.1. Simulation Results

As shown in Figure 7 and Figure 8, the Lamb waves propagation (displacement) in the CFRP–rubber bonding model with different sizes of interfacial debonding defects at the same location is observed from different angles. As shown in Figure 7, scattering clearly occurs when an incident wave passes through the debonding region (white solid line). This phenomenon becomes more prominent as the size of the debonding defect increases. The incident wave excites the A0 mode Lamb wave in the CFRP–rubber bonded plate, as shown in Figure 8. The wave passes through the debonding area (red circle) and continues to propagate forward in the CFRP sheet without further energy leakage into the rubber layer. The magnitude of the wave energy in the CFRP plate at the corresponding defect location is proportional to the size of the debonding defect. The wave energy at the observation point (pink circle) also increases with the size of the debonding defects in the bonded structure.

We extracted the time domain and envelope of the excitation signal and the received signal from the sensor in the simulation model, as shown in Figure 9. It should be noted here that the amplitude of the excitation signal is scaled to show more details of the received signal. The wave packet transmission time ($t_{g}$) extracted from the figure is 0.672 ms. The distance between the actuator and sensor is 325 mm. The calculated group velocity of the wave packet is 483.33 m/s.

According to Figure 4c, the analytical solution of the group velocity of the A0 mode at 100 kHz is 478 m/s. The maximum error between the simulation results and analytical solution is 1.1%, which can be considered as an acceptable value. It can be seen that the finite element method established in this study can correctly simulate the propagation of Lamb waves in CFRP–rubber bonded plates, and it is confirmed that the propagation in the plates is an A0 mode Lamb wave.

#### 3.2.2. Time-Domain Analysis (Simulation)

The first wave packet of the acquired signal was used to exclude the effect of boundary reflections. The acquired signals of the observation points at the same location in the CFRP–rubber bonding model with different sizes of interfacial debonding defects are shown in the time domain in Figure 10. Compared to the results without debonding (intact structure), the amplitudes of the other signals tend to increase with the size of the debonding defect. Similarly, the increasing scale of interface debonding damage produces damage detection time-domain signals with significant phase delays relative to the nondestructive detection time-domain signals.

#### 3.2.3. Frequency Domain Analysis (Simulation)

In addition to the time-domain analysis, signal analysis includes frequency and time-frequency domain analyses. In the time-domain analysis, information, such as the amplitude and phase of a signal, is often used as an indicator to identify structural damage. Frequency-domain analysis can be used to complement the time-domain analysis to further observe the signal characteristics based on frequency. Commonly, the time-domain signal is converted into a frequency-domain signal using a Fourier transform.

Fast Fourier transform was used to analyze the time-domain signal in Figure 10 in the frequency domain. Subsequently, the results were compared with those of the time-domain analysis. The results of the Fourier transform are shown in Figure 11, indicating the increased amplitude of the acquired signal with an increase in the debonding size, which is consistent with the results of the time-domain analysis and principle of interface debonding detection in Section 2.2.

#### 3.2.4. Simulation of the DEF Calculation Results

After performing the Fourier transform of the simulated time-domain signals, the amplitudes of the different interfacial bonding scales corresponding to 100 kHz frequency in Figure 10 were extracted. The DEFs for the different debonding scales in the simulation model were calculated using Equation (7), as shown in Figure 12. DEF exhibits an increasing trend with a gradual increase in the degree of interfacial debonding defects.

### 3.3. Experiments

Physical experiments were performed to validate the results of simulations. CFRP–rubber sheet specimens, replicating the simulation model in both material characteristics and bonding conditions, were meticulously prepared. The surface was polished with sandpaper to ensure the quality of the bonding [24]. An A/B epoxy adhesive mixed at a mass ratio of 1:1 was uniformly applied to the bonding area of the CFRP sheet (Toray T300 grade sheet produced by Jiangsu Boshi Carbon Fiber Technology Co., Ltd.: Nantong, China) with a brush, followed by placing the cut rubber sheet. After placing the CFRP side of the bonded structure horizontally facing upwards, a uniform pressure was applied to the CFRP sheet for two days to remove the excess adhesive and air bubbles and ensure a uniform bonding.

Interfacial debonding damage in the prepared bonded plates was achieved by placing polytetrafluoroethylene (PTFE) film sheets of different sizes with a thickness of 0.1 mm at specific locations [34]. The PTFE film is considered a nonstick coating owing to its resistance to various organic solvents; thus, it was embedded in the debonding area to simulate delamination damage [23]. Finally, the cured bonding plate was cleaned using 95% pure industrial alcohol to remove excess contaminants, such as the adhesive.

PZT sensors with a diameter of 2 cm were mounted at designated positions on the bonding element using a strong double-sided adhesive. A five-cycle sinusoidal signal modulated by a Hanning window was used as the incident signal input to a Tektronix AFG3252C (Tektronix Inc.: Beaverton, OR, USA) arbitrary function generator. The peak voltage of the input signal was amplified from 5 V to 100 V (Aigtek ATA-2021B: Model ATA-2021B high voltage amplifier from Xi’an Antai Electronic Technology Co., Ltd. (Aigtek), Dongguan, China). The amplified excitation signal was applied to the PZT piece as an actuator at the left end of the bonding member. The signals sensed by the PZT sensors were captured using a Tektronix DPO5104B (Tektronix Inc.: Beaverton, OR, USA) digital fluorescence oscilloscope and transferred to a host computer for postprocessing. The prepared test samples and experimental setup are shown in Figure 13 and Figure 14, respectively.

#### 3.3.1. Experimental Results

A continuous wavelet transform (CWT) was used to filter the noise signal and ensure the accurate extraction of the feature signals during the data analysis to reduce the influence of environmental noise on the detection results. CWT provides a complete representation of the signal by continuously changing the translation and scale parameters of the wavelet [35]. The signal was denoised using a Daubechies wavelet.

#### 3.3.2. Time-Domain Analysis (Experiment)

The experimental signals acquired by the CWT-processed actuator–sensor pair along the centerline are shown in Figure 15. The experimental time-domain signals exhibit a similar trend to that of the simulation results as the scale of the interfacial debonding defects changes.

#### 3.3.3. Frequency Domain Analysis (Experiment)

Similar to the frequency-domain analysis of the simulated data, Fourier transform was applied to the time-domain signal of the experimental test; the results are shown in Figure 16. The amplitude of the sensed signal increased as the interfacial debonding damage scale increased. These results were in good agreement with the simulation results.

#### 3.3.4. Experimental DEF Calculations

The amplitude values corresponding to different interfacial debonding scales at a frequency of 100 kHz after the Fourier transform were extracted from the experimental time-domain signals. The experimental DEF values for different debonding cases were calculated using Equation (7), and the results are shown in Figure 17. The DEF value increased with increasing interfacial debonding defects.

## 4. Discussion

The feasibility of the interfacial defect detection relying on the Lamb wave debonding energy loss was verified. The numerical and experimental data were processed, and the DEF increased with increasing damage levels, showing similar trends. However, some discrepancies remain, which can be attributed to the following reasons.

During the preparation of the test pieces, the adhesive inevitably spread, resulting in irregular boundaries in the debonding area, which are different from the simulation and theoretical analysis.The numerical simulations did not consider the influences of electromechanical coupling and bonding layers, simplifying the model. In the experiment, wave energy leaked into the bonding layer during propagation, which inevitably led to discrepancies between the calculated and simulated results.

As shown in Figure 11 and Figure 14, the time-domain signals in the numerical simulations and experiments demonstrate that the increase in the interface debonding damage scale generates damage detection time-domain signals with a significant phase delay relative to the nondestructive detection time-domain signals. This is ascribed to the different frequency dispersion characteristics of the waves propagating through the structure in areas with different interface damage states [22]. In this study, interface debonding defects were mainly determined by the change in the energy magnitude of the induced signal; the change in wave speed due to debonding defects was not discussed in depth.

## 5. Conclusions

In this study, we developed a novel method for detecting interfacial debonding defects in CFRP–viscoelastic (rubber) bonded structures by analyzing the energy loss of active Lamb waves. This method introduces a Damage Evolution Factor (DEF), which provides a quantitative measure of the debonding damage. The effectiveness of this method was validated through both numerical simulations and experimental tests, revealing several key findings: 

Method Effectiveness: The proposed detection method demonstrated strong effectiveness in identifying interfacial debonding defects, particularly by using a simple configuration that requires only a PZT sensor placed on the external surface of the structure. This makes the method highly suitable for integration into existing SRM structures with minimal operational complexity. Numerical and Experimental Validation: Numerical simulations showed that debonding defects at the interfaces caused significant wave scattering without energy leakage to the inner structure. The first arrival wave packet, extracted from the time-domain signal, was used for DEF analysis, with DEF values increasing proportionally to the debonding damage scale, confirming the reliability of this approach.DEF Application and Limitations: While DEF provides a qualitative assessment of debonding damage, uncertainties in parameters β and γ limit its quantitative application. Parameter β is related to the material’s elastic modulus and density, and γ to the structure’s thickness, shape, and boundary conditions. β can be estimated through tensile tests or material handbooks, while γ requires finite element simulation or theoretical analysis. Further research is needed to enhance the quantitative assessment of interfacial debonding damage in CFRP–viscoelastic bonded structures.Future Work: Future research should focus on refining the DEF metric to address uncertainties in material parameters and exploring advanced signal processing techniques to improve detection accuracy and robustness, particularly in complex bonded structures.

In conclusion, the proposed method and DEF metric offer a promising approach for detecting and evaluating interfacial debonding defects in bonded composite structures, with significant implications for the structural integrity and reliability of solid rocket motors during service and storage periods.

## Figures and Tables

**Figure 1 sensors-24-05567-f001:**
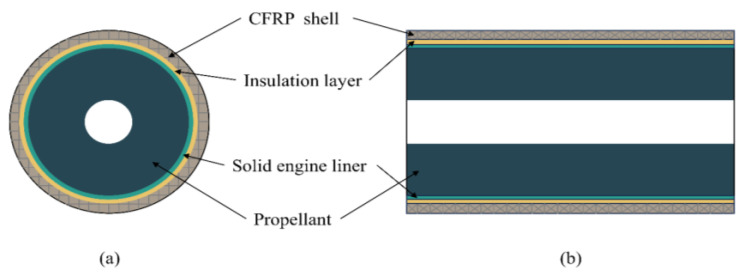
Schematic of a solid rocket motor: (**a**) radial section and (**b**) axial section.

**Figure 2 sensors-24-05567-f002:**
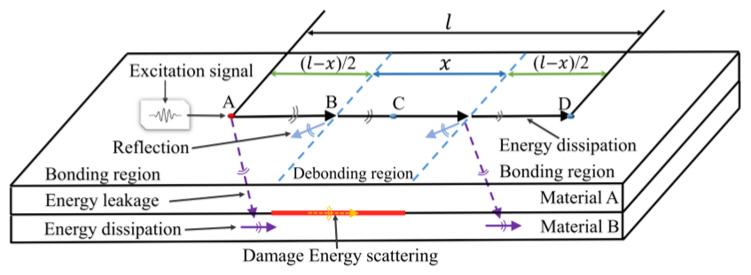
Schematic of the Lamb wave energy propagation for the interface debonding defect structure. Point A is actuator, point B as the left boundary of the debonding damage, point C and D are the sensors.

**Figure 3 sensors-24-05567-f003:**
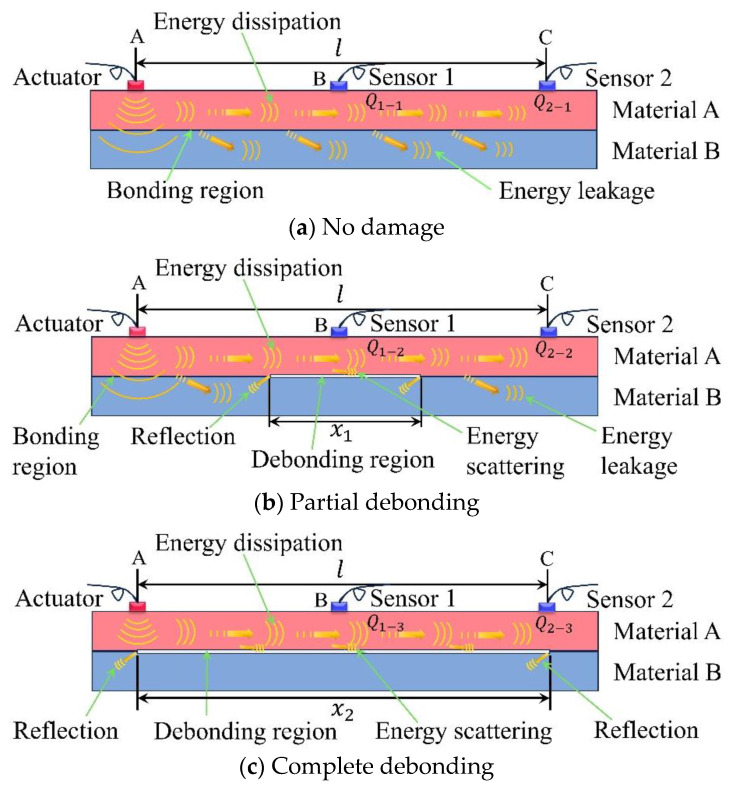
Detection method of the Lamb wave energy interface debonding based on a lead zirconate titanate (PZT) sensor.

**Figure 4 sensors-24-05567-f004:**
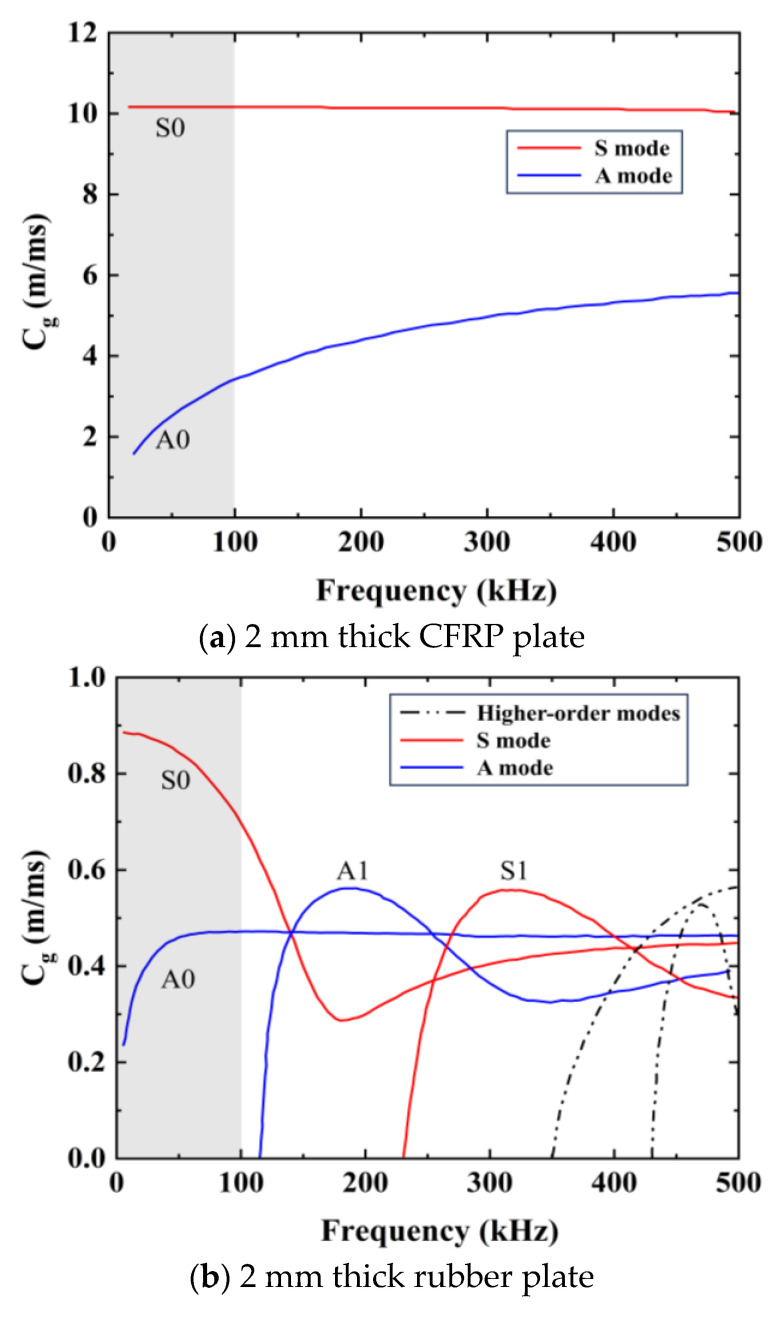
Dispersion curves (group velocity–frequency) of different structures.

**Figure 5 sensors-24-05567-f005:**
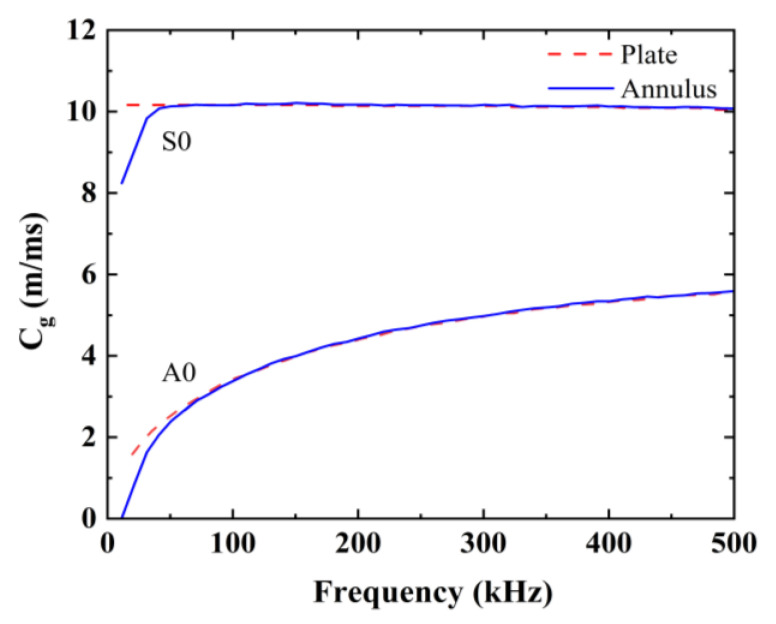
Dispersion curves for the flat plate and ring–space structures.

**Figure 6 sensors-24-05567-f006:**
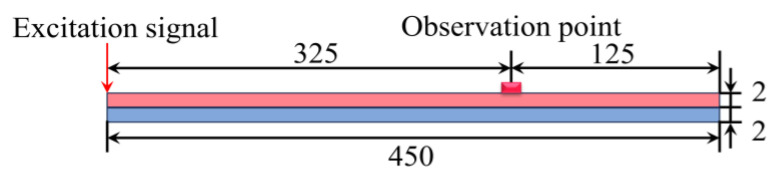
Dimensions of the carbon fiber-reinforced polymer (CFRP)–rubber bonding plate (mm).

**Figure 7 sensors-24-05567-f007:**
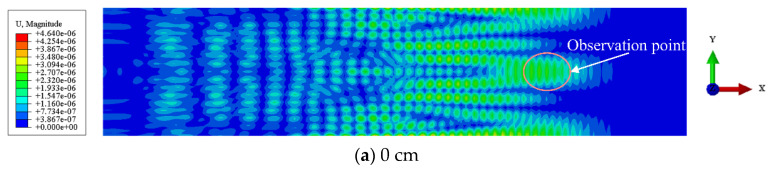
Top view of Lamb wave propagation (displacement) in the CFRP–rubber bonding model with different sizes of the interfacial debonding defects.

**Figure 8 sensors-24-05567-f008:**
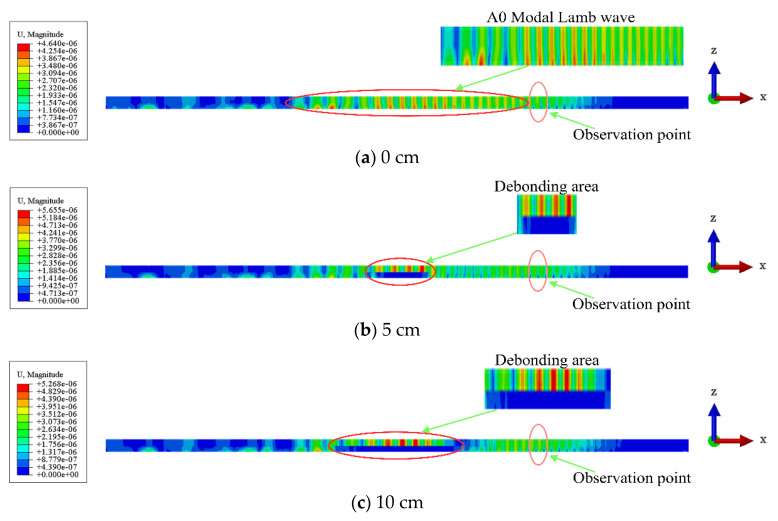
Main view of Lamb wave propagation (displacement) in the CFRP–rubber bonding model with different sizes of the interfacial debonding defects.

**Figure 9 sensors-24-05567-f009:**
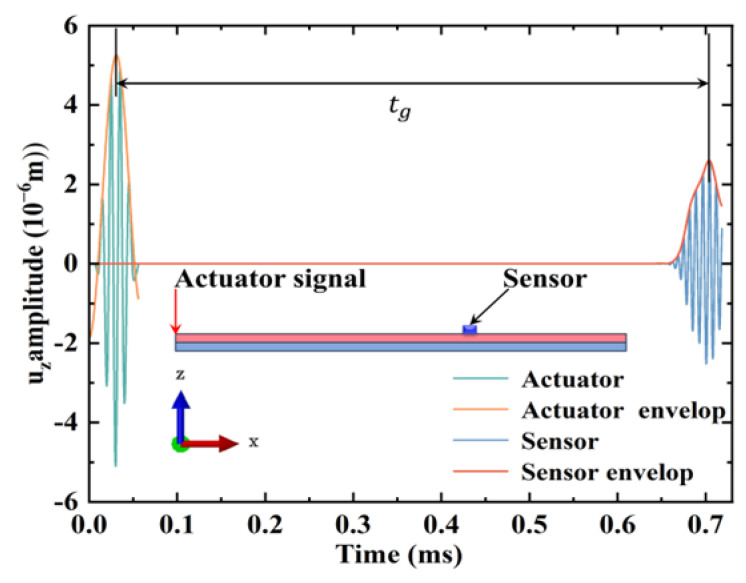
Group velocity calculation diagram.

**Figure 10 sensors-24-05567-f010:**
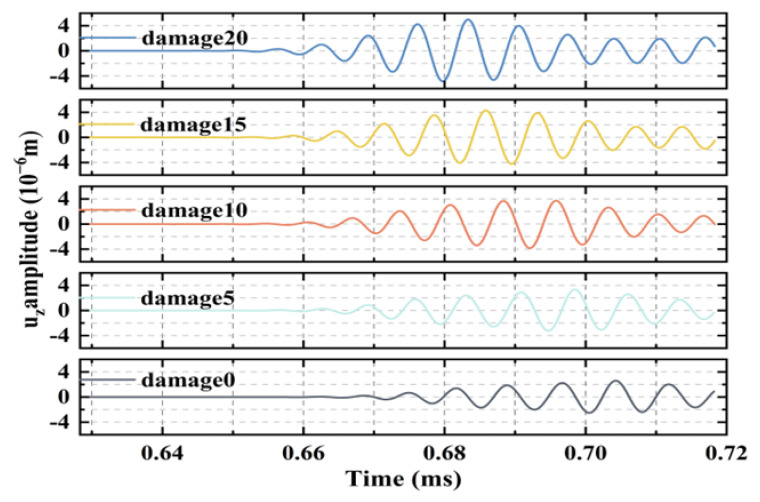
Simulation results of the time-domain signals for different debonding defects.

**Figure 11 sensors-24-05567-f011:**
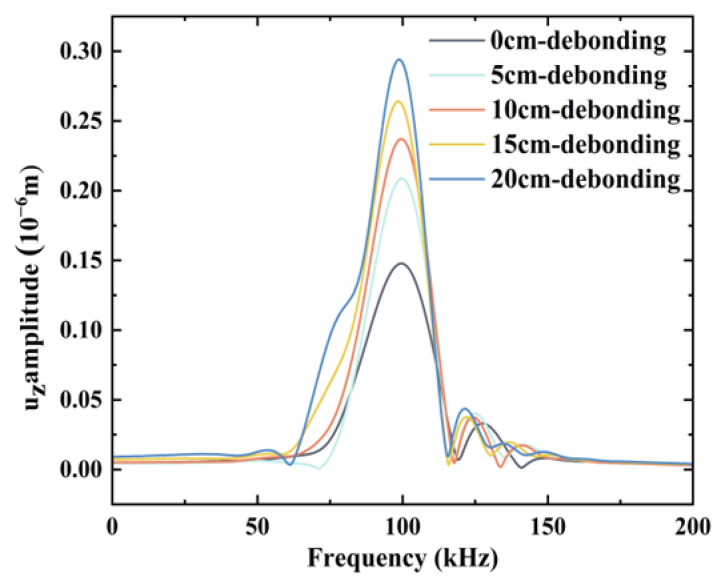
Fourier transform results of the simulated time-domain signal.

**Figure 12 sensors-24-05567-f012:**
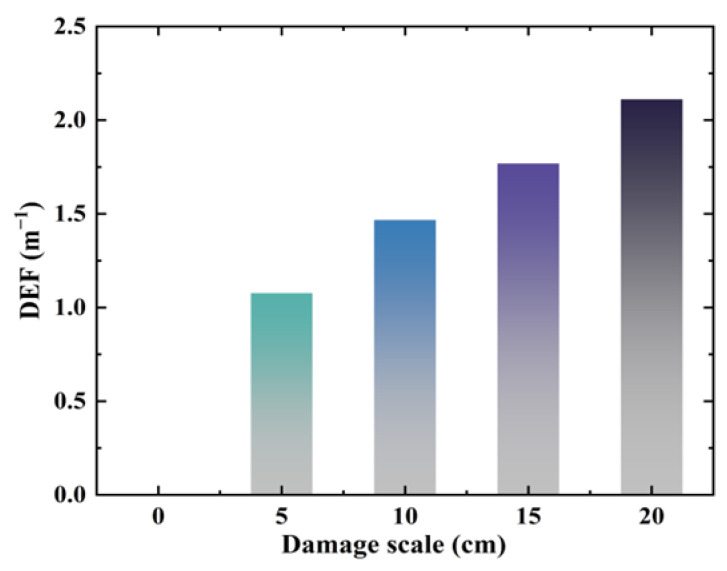
Simulation DEF calculation results.

**Figure 13 sensors-24-05567-f013:**
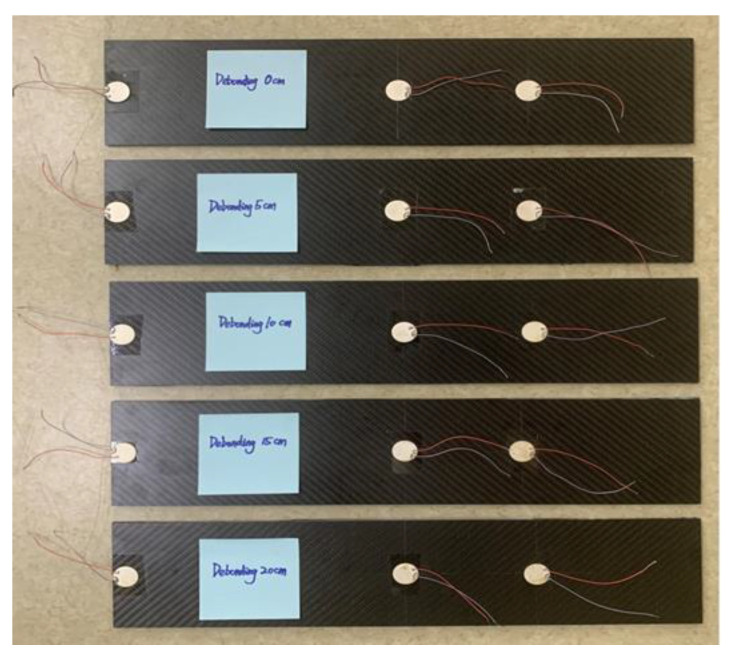
Preparation of the multidamage test samples.

**Figure 14 sensors-24-05567-f014:**
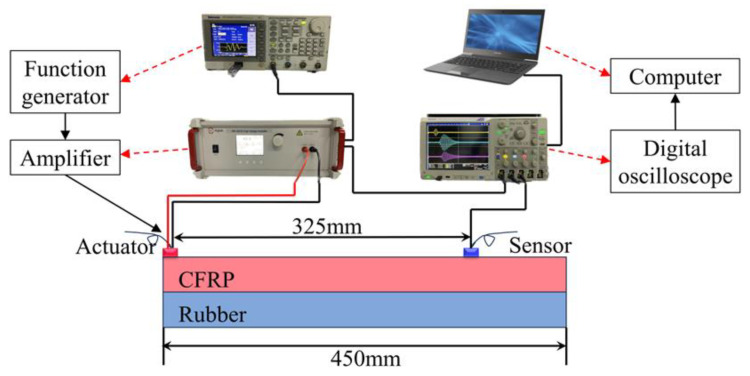
Experimental platform.

**Figure 15 sensors-24-05567-f015:**
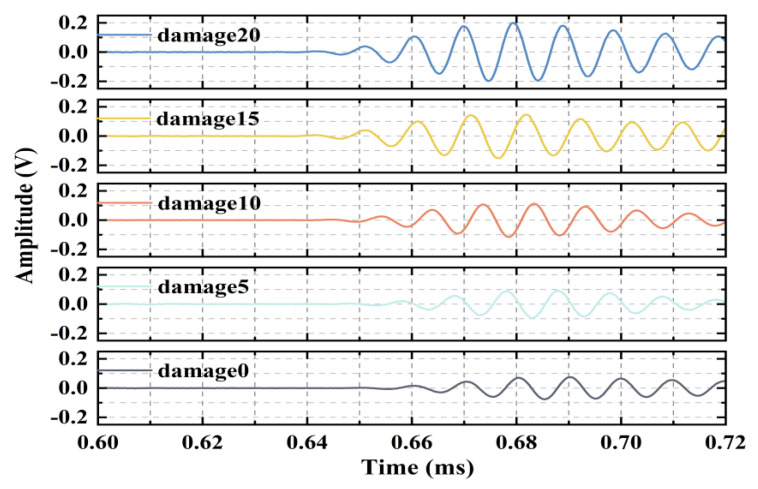
Experimental results of the time-domain signals for different debonding defects.

**Figure 16 sensors-24-05567-f016:**
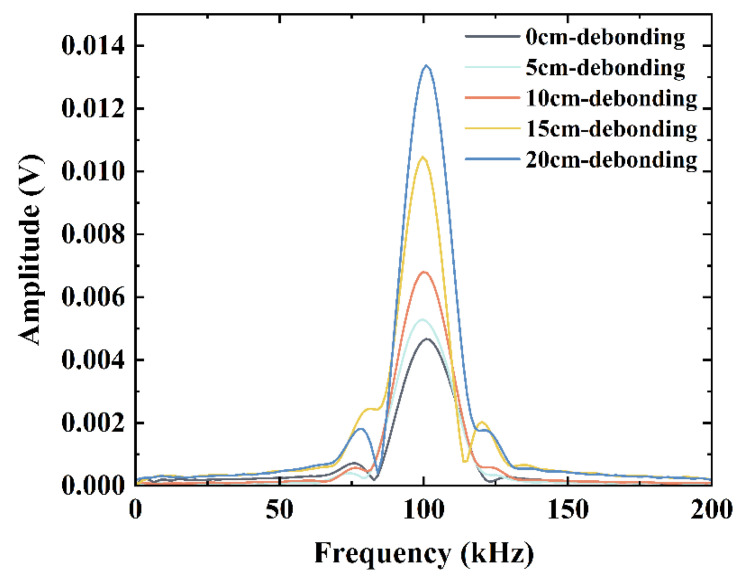
Experimental time-domain signal Fourier transform results.

**Figure 17 sensors-24-05567-f017:**
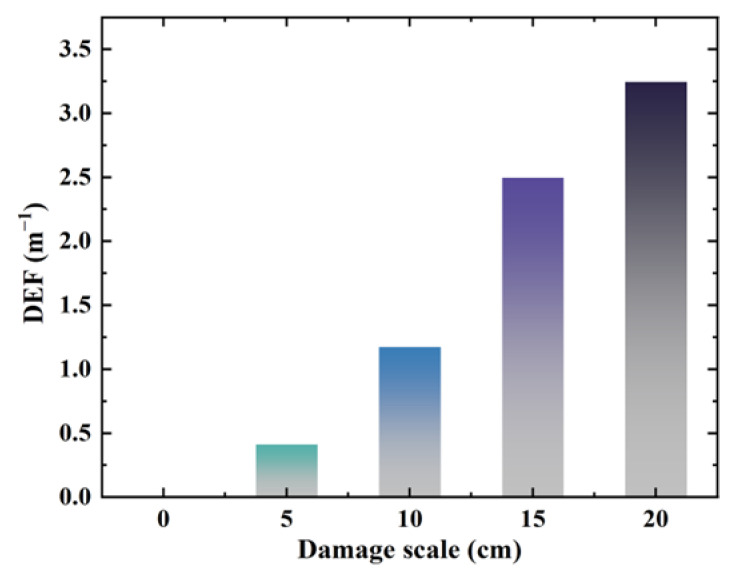
Experimental DEF calculation results.

**Table 1 sensors-24-05567-t001:** Material parameters.

Material Properties	Density (kg/m^3^)	Young’s Modulus (GPa)	Poisson’s Ratio	Thickness (mm)
CFRP	1900	177	0.32	2
Rubber (viscoelastic)	1150	0.7145 + 0.3487*i*	0.4614 − 0.0214*i*	2

## Data Availability

The data utilized in this study are available from the first or corresponding authors upon a reasonable request.

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
