# Peer review of "Detection of Debonding Defects in Carbon Fiber-Reinforced Polymer (CFRP)–Rubber Bonded Structures Based on Active Lamb Wave Energy Analysis"

_sensors, 2024, doi:10.3390/s24175567_

Round 1
Reviewer 1 Report
Comments and Suggestions for Authors
This paper presents a study on a methods of interfacial debonding defects in CFRP-rubber bonded structures based on active lamb waves energy analysis. However, it seems the manuscript is not well prepared as there are many editorial mistakes. The following comments are made for the authors to improve the manuscript:
1. In Table 1. the Young’s modulus and Poisson ratio are expressed as 0.7145+0.3487i and 0.4614-0.0214i. However, the meaning of i has never been explained in the text.
2. In Figure 4, the meaning of Cg is not explained in the text. Besides, there is no legend for this figure.
3. In Figure 5, there are four lines with only two legends.
4. The captions of Figures 7 and 8 indicate both figures are to describe the stress distribution of the wave propagation. However, the legend in the figure shows displacement (i.e., U, magnitude).
5. Figures 9 and 10 describe the same response. However, the title of the y-axis of Figures 9 and 10 are different. Besides, there are too many curves in both figures that using different colors only is not enough to distinguish them.
6. Figure 11 describes the DEI which is different from “DEF” in the section title.
7. In the experiment section, the authors describe the amplitude of voltage, which is different to the amplitude (deformation) in the previous section. Please clarify the reason of using different parameters in the comparison.
8. Iine 188-189, please clarify how to determine parameters β and γ.
9. Figure 6 is confusing. Please add an annotation of the observation point as shown in Figure 7 to Figure 6. Meanwhile, please add annotations of the two sensors shown in Figure 6 to Figure 7. Same comments applies to Figure 8.
10. In reality, how to position the sensors given the debonding positions are unkown.
Reviewer 2 Report
Comments and Suggestions for Authors
Comments:
This study explores the detection of interfacial debonding defects in CFRP-viscoelastic bonded structures utilizing Lamb wave energy loss. The research addresses the critical issue of maintaining the structural integrity of solid rocket motors, underlining the necessity for advanced nondestructive testing techniques. A novel methodology based on the energy dissipation of Lamb waves is proposed, with the introduction of a debonding evaluation factor (DEF) to quantitatively assess the extent of interfacial debonding. Comprehensive theoretical derivations, finite element simulations, and experimental validations are performed to evaluate the efficacy and precision of this approach. However, the following issues are suggested to be well addressed to improve the manuscript quality.
1. In the introduction, the author extensively discusses various methods, including X-ray and infrared thermography, and summarizes their limitations. However, the advantages of the Lamb wave method utilizing PZT sensors are not specifically addressed.
2. The experimental results offer a basic validation of the simulation content, predominantly presenting a qualitative description. They merely reflect a trend without demonstrating the quantitative capabilities of Lamb waves.
3. Do the experimental results exhibit repeatability, and is the accuracy of these results influenced by the variations in material specimens as well as the precision in the location and size of the embedded damage?
4. There are several typos and vague sentences in the manuscript, so the manuscript should be carefully checked and thoroughly modified.
5. In the context of the multilayer structure, was the effect of the A/B epoxy adhesive considered in the plotting of the group velocity images?
6. Line 211 mentions the use of "GUIGUW" software to plot dispersion curves for different materials. For multilayer structures and structures with adhesive layers, is it possible to use this software to plot the overall dispersion curves of the entire structure, rather than plotting them separately or disregarding the effects of the adhesive layers?
7. In line 257, the "Tie" constraint is employed to simulate the adhesive layer between CFRP and rubber. Has a comparative analysis been conducted to evaluate the effectiveness of other simulation methods?
8. In Figure 8, how is the A0 mode of the Lamb wave identified?
9. In Section 3.3.2, can the signal analysis incorporate a description of the phase variation corresponding to the progression of damage?
10. What is DEI in Figure 11?
Comments on the Quality of English Language
There are several typos and vague sentences in the manuscript, so the manuscript should be carefully checked and thoroughly modified.
Reviewer 3 Report
Comments and Suggestions for Authors
Thank you very much for your work on the paper.
Unfortunately, I have to come to the conclusion that it offers little to no new scientific value.
The effect described has been known in the literature for decades and has been investigated in various papers. In addition, the measurement effect is obvious if you have a basic knowledge of wave physics.
The paper is a nice study, but should be seen as a conference contribution rather than a journal paper.
In addition, the following comments on the content:
p. 1, line 24: The introduction is very superficial and poorly researched. The authors claim that essential studies (e.g. debonding in CFRP, CFRP with viscoelastic media, etc.) do not or hardly exist. This is not true. Furthermore, it is not entirely clear whether an NDT or an SHM method is to be developed. From the description, it looks more like an NDT method using Lamb waves. The use of Lamb waves does not mean that it is an SHM method.
p. 2, line 85: PZT is not the abbreviation for piezoelectric. PZT stands for lead zirconium titanate. The ceramic base material of the piezoelectric sensor. If the authors use an abbreviation, please use the correct one.
p. 2, line 85: The authors confuse the terms here. The use of guided waves in the ultrasonic range is not automatically SHM. The guided waves are the basis of the physical measuring principle. The difference between NDT and SHM is another one. The authors describe an NDT method, an inspection or test, rather than permanent monitoring.
p. 3, line 90: That is a very gross understatement. SHM using Lamb waves in particular has been studied and researched for decades in the characterisation of CFRP. This is particularly true in the aerospace and automotive sectors. The reviewer recommends that the authors research at least 3 to 5 sources on this topic in order to have an overview of the subject.
p. 3, line 93: That's not true either. This topic in particular is often investigated and is of great interest.
Please check for example:
Orta, Adil Han: Inverse characterization of the orthotropic and viscoelastic material properties of lightweight plates using full field guided wave propagation data; 2023, KU Leuven, KU Leuven
Huber, Armin M. A.: Classification of solutions for guided waves in fluid-loaded viscoelastic composites with large numbers of layers; 2023, The Journal of the Acoustical Society of America , Vol. 154, p. 1073-1094
p. 5, line 167: Do not split a Figure on 2 pages. Please put Figure 3 on only one page.
p. 5, line 171: the (not with capital letter)
p. 6, line 195: This description seems to fit what the authors are doing. As mentioned in the introduction, the authors do not necessarily apply SHM just because Lamb waves are used. The terms used throughout the paper should be reviewed and adjusted in this regard
p. 7, line 222: A legend is missing. S0 and A0 are clear due to the colors. But what about the dotted lines?
p. 7, line 223: Again, please do not split a figure on 2 pages.
p. 8, line 250: Without knowing the structure of the laminate, this is an incomprehensible restriction. The most important question is: what is the layer structure of the composite? Table 1 already gives averaged values. Are these values for one layer? What is the layer structure?
p. 8, line 264: This is where the assumption of isotropy comes into play. The anisotropy of the composite has an influence in the x-y plane. This is completely ignored. Once again, without knowing the layer structure, this approximation cannot be understood. Nevertheless, it can be assumed that it will not have a major influence on the results.
p. 15, line 408: That is correct. With a large debonding area, much of the energy will spread out in a thinner section of the plate. This changes the group velocity. However, with a skilfully chosen time window in the time signal, this effect will hardly have any impact.
p. 15, line 414: Both the simulation and the experiment are kept very simple. The design of the experimental set-up provides many starting points for making the process even rudimentarily usable.
p. 15, line 417: Laboratory equipment is used in the experiments. How do the authors visualise the measurement in reality? What hardware should be used?
p. 16, line 424: Both the experiment and the laboratory test provide for debonding in the direct path between the actuator and the sensor. This cannot be guaranteed in reality, as the location of the debonding is not known. This is the most important point of further investigations: How does the process behave if the debonding is not in the direct path between actuator and sensor? This is where the approximation of isotropy comes into play and cannot be utilised.
Round 2
Reviewer 1 Report
Comments and Suggestions for Authors
The authors haven't addressed all the comments provided in the first round of review. For example, The meaning of Cg is not explained in the manuscript though it is well explained in the response. The captions of Figures 7 and 8 of the revised manuscript indicate the stress distribution which is not consistent with the legend. For the comment 8, the determination process shall be introduced in the revised manuscript but not only in the response. Because in the end, the readers cannot read the response. The comment 9 is not addressed regarding adding annotations to Figure 7.
Reviewer 3 Report
Comments and Suggestions for Authors
Firstly, the authors have taken all the comments to heart and revised them. In particular, comments regarding terminology and literature have been taken into account. The use of terms is now clearer and more consistent. The state of the art has been more analysed and presented.
Moreover, comments on formatting have been taken into account and implemented. Additional information on the experimental procedure has been added and makes the description of the work more complete.
All in all, however, my summary remains that the paper offers little scientific novelty. The effect described (energy loss of a Lamb wave mode when interacting with a delamination) is well known and has often been investigated.
Nevertheless, the authors have explained the necessity of developing this NDT technique. From an engineering point of view, this is certainly an interesting topic and the development of the measurement method is necessary. But from a scientific point of view, it offers no new value.
